# Quantification of Pre-Season and In-Season Training Intensity across an Entire Competitive Season of Asian Professional Soccer Players

**DOI:** 10.3390/healthcare10081367

**Published:** 2022-07-23

**Authors:** Hadi Nobari, Akhilesh Kumar Ramachandran, João Paulo Brito, Rafael Oliveira

**Affiliations:** 1Department of Motor Performance, Faculty of Physical Education and Mountain Sports, Transilvania University of Braşov, 500068 Braşov, Romania; 2HEME Research Group, Faculty of Sport Sciences, University of Extremadura, 10003 Cáceres, Spain; 3Sports Scientist, Sepahan Football Club, 81887-78473 Isfahan, Iran; 4Sports Dynamix Private Limited, Chennai 600006, India; akhil24.kumar@gmail.com; 5Sports Science School of Rio Maior—Polytechnic Institute of Santarém, 2040-413 Rio Maior, Portugal; jbrito@esdrm.ipsantarem.pt; 6Research Centre in Sport Sciences, Health Sciences and Human Development, Quinta de Prados, Edifício Ciências de Desporto, 5001-801 Vila Real, Portugal; 7Life Quality Research Centre, 2040-413 Rio Maior, Portugal

**Keywords:** heart rate, high-speed running, monitoring, sprint, sports technology, training load, load quantification, competition phases

## Abstract

The aim of this study was to quantify the training load in two microcycles (Ms) from pre- and another two from in-season and to analyze playing position influences on the load experienced by professional soccer players. Nineteen Asian athletes, including four central defenders, four wide defenders, six central midfielders, three wide midfielders, and two strikers participated in this study. The micro-electromechanical system was used to collect training duration, total distance, and data from Zone 1 (0–3.9 km·h^−1^), Zone 2 (4–7.1 km·h^−1^), Zone 3 (7.2–14.3 km·h^−1^), Zone 4 (14.4–19.7 km·h^−1^), and Zone 5 (>19.8 km·h^−1^), heart rate maximum (HRmax), and average (HRavg). The load was reduced on the last day of the Ms, with the exception of Zone 5, in M1, where higher values were found on the last day. Significant differences were observed between central and wide defenders for distance covered in Zone 4 (effect-size: ES = −4.83) in M2 and M4 (ES = 4.96). Throughout all the Ms, a constant HRmax (165–188 bpm) and HRavg (119–145 bpm) were observed. There was a tendency to decrease the load on the last day of the Ms. In general, there were higher external training loads in Ms from the pre-season than in-season. Wide defenders and wide midfielders showed higher distances covered with high-intensity running.

## 1. Introduction

The increasing intensity of soccer matches has resulted in drastic changes in the physiological demands of the sport [1,2]. The soccer game involves brief bouts of linear high-intensity running, sprinting, accelerations, decelerations, and multi-directional activities or change-of-direction movements, interspersed by longer recovery periods of lower intensity activities [3]. For instance, players in the English Premier League have been found to cover 681 m at running speeds ranging from 19.5 to 25.1 km·h^−1^ [4]. Previously, professional soccer players have performed 150–200 intense actions per game [5] and covered very high-intensity distances every 72 s [6]. The findings from the available literature highlight the importance for players to maintain high physical fitness levels to withstand the increasing physical demands during the training session and competitive matches [7]. Therefore, monitoring/tracking each player’s daily intensity has become a necessity for coaches and practitioners in order to maintain an optimal fitness level of the players throughout the competitive season [8], and also to reduce the risk of injuries [9].

The weekly intensity of soccer players varies according to the number of matches during the season and phase of the annual plan [10,11]. The aim during the pre-season is to rebuild the fitness levels of the players after the off-season [12]. However, the aim during the in-season is to maintain and if possible improve the specific fitness components developed during the pre-season [12]. Professional soccer players train between four to six sessions/week during the competitive season [10], while it might be as high as one or two sessions, five days a week during the pre-season [11]. Such variation in training patterns might drastically influence the physiological demands placed on players and might result in differences in the physiological stress [13]. Therefore, knowledge regarding the intensity, during the different phases of the season, can be helpful for coaches in designing and implementing periodized soccer-specific training routines.

In this sense, load monitoring includes both internal and external dimensions. For better clarity, we avoid the load term and replace it with intensity following a recent suggestion by Staunton et al. [14]. Thus, the external dimension constitutes the physical demands performed during a training session or match, while the internal dimension is related to biochemical and biomechanical responses to the external stressors [15]. In soccer, subjective scales such as the rating of perceived exertion (RPE) initially developed by the Borg scale [16] and later adapted by Foster [17], and s-RPE developed by Foster et al. [18,19] and Impellizzeri et al. [20] were initially used to quantify internal intensity. This was followed by using heart rate (HR) telemetry to monitor the cardiovascular response during each training session. On the other hand, the external intensity is measured in training and matches using micro-electro-mechanical systems (MEMS) such as location position systems, global positioning systems (GPS), or optical camera tracking systems, using metrics such as distance covered and intensities of different movements/actions [21]. Therefore, soccer teams in recent years employ a combination of the above-mentioned methods to quantify external and internal intensities.

Recent research in season intensity quantification has provided valuable information by players in various leagues across European soccer [22,23,24,25,26,27]. Further, several studies have attempted to quantify the in-season training intensity in a professional soccer team, which involves a comparison of training days within weekly mesocycles [26,27,28]. Based on the evidence available from the existing literature, the intensity has shown to have little variation during the competition phase [26,27]. Additionally, within the weekly microcycles (Ms), the intensity was generally similar within training days with a reduction in the intensity in the days prior to the competition [22,24,26]. However, a comparison of pre-season and in-season has been made by few studies [12,29]. Thereafter, there is still paucity in the available evidence, since these studies have only reported limited information on intensity and have considered only the duration of training and s-RPE for the internal intensity quantification, and not the external intensity data collected using MEMS. Additionally, there are currently not many available studies that have analyzed internal and external intensities at two different time points during the soccer pre- and in-season of the Asian Professional League.

A better understanding of the periodization practices in professional soccer will be helpful for performance and technical coaches to manipulate the training volume, in order to help players increase their physical levels during different phases of a soccer season [30]. However, due to the lack of clear evidence in professional players, the periodization practices in soccer are currently poorly known and are reliant on the coach’s training philosophy based on years of coaching experience [31]. Therefore, it can be argued that there is still a lack of clarity with respect to periodization practices and whether these practices demonstrate the required variation in intensity. Since the coaching practices vary, further research is needed to broaden our understanding of this topic, and how it should be programmed across an entire season. Moreover, most of the research on intensity during training and competition during various Ms have predominantly focused on European soccer clubs [22,24]. As a result, these findings cannot be applied across all the soccer teams, mainly due to the variation in match demands, culture, the number of matches played, and tactical reasons [32]. To the author’s knowledge, there are limited studies [32] that have quantified intensity from both soccer pre- and in-season in Asian Professional League. Therefore, the primary aim of this study was to quantify intensity in four different Ms from the pre- and in-season period (two Ms from each period) of soccer players competing in an Asian Professional League. The secondary aims of the study were to compare training days in each M as well as to compare playing positions.

## 2. Materials and Methods

### 2.1. Experimental Approach to the Problem

For this longitudinal study, training data were collected over a 48-week period during full season. For the purposes of the present study, all training sessions conducted during the main team sessions were considered which means that rehabilitation or recovery sessions were not used for analysis. The duration of the training sessions includes the warm-up, main and slow-down phases, plus stretching. All training programs were planned by the coach and staff, and the researchers only standardized the first 30 min and the final 30 min, (i.e., before and after each training session) regarding research procedures for external and internal data collection (described below in Section 2.4 and Section 2.5).

The weeks were chosen based on similar characteristics in pre- and in-season periods which included pre- and in-season weeks with the exact same scheduling, (i.e., M1 and M4 had five training sessions while M2 and M3 had four training sessions). Such similarities would allow a proper comparison between accumulated data. The characterization of weekly training Ms analyzed in the present study is described in Table 1.

### 2.2. Participants

Nineteen professional soccer players belonging to a soccer team in Asian Professional League participated in this study. The players were four central defenders (CD), four wide defenders (WD), six central midfielders (CM), three wide midfielders (WM), and two strikers (ST). They had an age of 28.4 ± 3.7 years, a height of 180.9 ± 7.1 cm, a body mass of 74.0 ± 7.9 kg, and a body mass index of 22.5 ± 1.1 kg/m^2^.

All players participated in the full season (>60% of the total matches played in a season). Specifically, for the present study, the inclusion criteria were a training session participation of 100% regarding frequency and duration of the session while the exclusion criteria were players who participated in rehabilitation or recovery sessions. All participants were familiarized with the training protocols prior to investigation. This study was conducted according to the requirements of the Declaration of Helsinki and was approved by the University of Isfahan research ethics committee (IR.UI.REC., 1399.064 Isfahan, Iran).

### 2.3. Procedures

All training sessions were conducted on the natural grass field and in the afternoon. The players had the same nutrition every week for at least several days in camp and throughout the match days. All the information was provided by the researcher and global positioning system (GPS) manager of the team who had more than eight years of experience. Training data were collected over the course of four different 7-day Ms: from pre-season, M1 and M2 included five and six training sessions, respectively; from in-season, M3 and M4 included six and five training sessions, with one match (national league), respectively. All Ms analyzed included consecutively training sessions. Days off or matches only occurred on the last days or the first day of the week (Table 1). Although other weeks also fit the descriptions provided, all weeks selected met the criteria for participants, meaning that they completed all training sessions during the chosen weeks. A total number of 22 training sessions (418 individuals) were observed for this study. This study did not influence or alter the training sessions in any way. Training data were collected at the soccer club’s outdoor training pitches. Data were analyzed per day of the week, (i.e., day 1, day 2, …,and day 7).

### 2.4. External Intensity Monitoring—Training Data

A 15 Hz MEMS (GPSPORTS systems Pty Ltd., model: SPI high-performance units, Australia) system was used in all the training sessions. This MEMS presents a 100 Hz accelerometer, with 16 G Tri-Axial-Track impacts, accelerations, and decelerations and 50 Hz Tri-Axial magnetometers. The GPS size dimensions were (74 mm × 42 mm × 16 mm). Additionally, it is water-resistant and uses infra-red, and weighed 56 g for data transmission. The validity and reliability of the device have been confirmed by Tessaro and Williams [33] study. Thus, the following variables were selected: total duration of training session, total distance, and distance of different exercise intensity zones which were adapted from previous studies [22,34]: Zone 1 (0–3.9 km·h^−1^), Zone 2 (4–7.1 km·h^−1^), Zone 3 (7.2–14.3 km·h^−1^), Zone 4 (14.4–19.7 km·h^−1^) and Zone 5 (>19.8 km·h^−1^). Considering that several players did not achieve a higher speed of 24 km·h^−1^ (which was Zone 6), Zone 5 included all data to avoid zeros and a lower comparison power analysis between players. Weekly zones (1–5) and weekly total distance were calculated by the sum value for the entire week for each M, respectively. 

### 2.5. Internal Intensity Monitoring—Training Data

A flashing red light was used to track HR. We placed each unit perpendicular to the bag and made sure the logo on the unit was facing backward and the RED light was on. High-performance units are designed to automatically collect athletes’ accelerometer and HR data in one session. In addition to the GPS receiver, the SPI Pro X unit consists of a tri-axial accelerometer, to estimate the forces on the player and an integrated HR monitor. The following variables were selected: maximal HR (HRmax) and average HR (HRavg). Then, weekly HRmax (wHRmax) and weekly HRavg (wHRavg) were calculated by the average value for the entire week for each M, respectively. The way this information was recorded was similar to previous studies [35,36,37,38]. 

### 2.6. Statistical Analysis

Data were analyzed using the IBM SPSS Statistics for Windows, Version 22.0. (IBM Corp., Armonk, NY, USA) statistical software package. Initially, descriptive statistics were used to characterize the sample. Shapiro-Wilk and Levene tests were conducted to determine normality and homoscedasticity, respectively. Once variables obtained a normal distribution (Shapiro–Wilk>0.05), it was used a repeated measures ANOVA test and the Bonferroni post hoc correction test to compare variables for days of the week. This process was repeated to also allow a comparison between all Ms/weeks and player positions. The results are significant with a *p* ≤ 0.05. Hedge’s g effect size (ES) was also calculated to determine the magnitude of pairwise comparisons. The following criteria in absolute values was used: <0.2 = trivial, 0.2 to 0.6 = small effect, 0.7 to 1.2 = moderate effect, 1.3 to 2.0 = large effect, and >2.0 = very large [39].

Considering the main of this study and using G-Power [40], a post hoc calculation was performed for *n* = 19, *p* = 0.05, ES = 0.6, one group, four measurements (four Ms), for repeated measures ANOVA which displayed 99.9% of actual power.

## 3. Results

### 3.1. Day-to-Day Intensity Variations across the Four Ms

Training duration is presented in Table 2 while Table 3 and Table 4 showed comparisons between days of the week for each M, respectively (all, *p* < 0.05). 

#### 3.1.1. M1, Pre-Season 

Total distance showed significant differences in day 1 > day 2 (ES = 6.40) and <day 3 (ES = −6.87). Day 2 < day 3 (ES = −12.10) and <day 4 (ES = −6.44). Day 3 > day 4 (ES = 8.25) and >day 5 (ES = 5.25). Additionally, day 4 < day 5 (ES = −3.39). 

Zone 1 showed significant differences in day 1 < day 3 (ES = −6.07), <day 4 (ES = −7.60), <day 5 (ES = −9.60). Day 2 < day 3 (ES = −4.82) and day 4 (ES = −6.92). Additionally, day 3 < day 5 (ES = −4.41). 

Zone 2 showed significant differences in day 1 > day 2 (ES = 6.13), <day 3 (ES = −8.62). Day 2 < day 3 (ES = −11.67) and <day 5 (ES = −0.66). Day 3 > day 4 (ES = 9.53) and day (ES = 7.99).

Zone 3 showed significant differences in day 1 > day 2 (ES = 7.71), >day 4 (ES = 11.71), >day 5 (ES = 7.01). Day 2 < day 3 (ES = −5.63). Day 3 > day 4 (ES = 8.64) and >day 5 (ES = 5.16). Day 4 < day 5 (ES = −3.31).

Zone 4 showed significant differences in day 1 > day 2 (ES = 3.54). Day 2 < day 3 (ES = −7.77) and <day 4 (ES = −4.45). Day 3 > day 5 (ES = 6.07). Day 4 > day 6 (ES = 3.34).

HRavg showed significant differences in day 1 > day 5 (ES = 2.35) and day 3 > day 5 (ES = 3.16).

#### 3.1.2. M2, Pre-Season

Total distance showed significant differences in day 1 < day 3 (ES = −16.63) and <day 5 (ES = −10.07). Day 2 < day 3 (ES = −10.47) and <day 5 (ES = −5.42). Day 3 > day 4 (ES = 17.59) and >day 6 (ES = 18.71). Additionally, day 4 < day 5 (ES = −9.08). 

Zone 1 showed significant differences in day 1 < day 2 (ES = −7.71), <day 3 (ES = −12.57), <day 5 (ES = −9.78), <day 6 (ES = −7.81). Day 2 < day 3 (ES = −4.26) and >day 4 (ES = 4.82). Additionally, day 3 > day 4 (ES = 10.56). Additionally, day 4 > day 5 (ES = −7.51) and >day 6 (ES = −5.12).

Zone 2 showed significant differences in day 1 < day 2 (ES = −11.26), <day 4 (ES = −7.03). Day 2 < day 3 (ES = −12.80), <day 4 (ES = −5.21) and <day 5 (ES = −7.31). Day 3 > day 4 (ES = 7.40) and >day 6 (ES = 15.79). Day 4 and day 5 were higher than day 6 (ES = 7.34, ES = 9.12, respectively).

Zone 3 showed significant differences in day 1 < day 2 (ES = −6.66), <day 3 (ES = −11.18), <day 5 (ES = −5.07). Day 2 < day 3 (ES = −4.01). Day 3 < day 4 (ES = 9.53), <day 5 (ES = 7.64) and <day 6 (ES = 9.86).

Zone 4 showed significant differences in day 1 > day 4 (ES = 5.92), <day 5 (ES = −6.42), >day 6 (ES = 7.04). Day 2 < day 3 (ES = −3.51), >day 4 (ES = 5.50), <day 5 (ES = −21.88), >day 6 (ES = 7.86). Additionally, day 3 > day 4 (ES = 8.55), <day 5 (ES = −13.78), >day 6 (E = 10.80). Additionally, day 4 < day 5 (ES = 1.99). Additionally, day 5 > day 6 (ES = 40.96).

Zone 5 showed significant differences in day 1 > day 3 (ES = −8.44), >day 4 (ES = 6.24), >day 6 (ES = 5.60). Day 2 < day 3 (ES = −14.28). Additionally, day 3 > day 4 (ES = 19.61), >day 5 (ES = 10.55), >day 6 (ES = 18.95).

HRmax showed significant differences in day 5 > day 6 (ES = 3.00). HRavg showed significant differences in day 2 > day 4 (ES = 3.67) and >day 6 (ES = 5.49). Additionally, day 3 > day 4 (ES = 4.00) and >day 6 (ES = 5.88). Finally, day 5 > day 6 (ES = 5.50).

#### 3.1.3. M3, In-Season

Total distance showed significant differences in day 1 < day 3 (ES = −5.46). Day 2 > day 4 (ES = 9.79), >day 5 (ES = 9.51) and >day 6 (ES = 13.57). Day 3 > day 4 (ES = 9.78), >day 5 (ES = 10.07) and >day 6 (ES = 11.40). Additionally, day 4 > day 6 (ES = 5.42). 

Zone 1 showed significant differences in day 1 < day 2 (ES = −5.55), <day 3 (ES = −6.55). Day 2 > day 4 (ES = 9.91), >day 5 (ES = 5.57) and >day 6 (ES = 12.09). Additionally, day 3 > day 4 (ES = 8.97), >day 5 (ES = 6.56) and >day 6 (ES = 10.30). Additionally, day 4 > day 6 (ES = 3.60).

Zone 2 showed significant differences in day 2 > day 4 (ES = 11.53), >day 5 (ES = 10.54) and >day 6 (ES = 9.18). Day 3 > day 4 (ES = 9.56), >day 5 (ES = 8.86) and >day 6 (ES = 7.36). Day 4 and day 5 were higher than day 6 (ES = −7.52, ES = −4.42, respectively).

Zone 3 showed significant differences in day 1 < day 3 (ES = −5.64). Day 2 < day 3 (ES = −8.30) and >day 6 (ES = 5.12). Day 3 > day 4 (ES = 9.02), >day 5 (ES = 11.47) and >day 6 (ES = 11.90). Day 4 > day 5 (ES = 7.79) and >day 6 (ES = 8.98).

Zone 4 showed significant differences in day 2 < day 3 (ES = −3.71), >day 4 (ES = 2.88), >day 5 (ES = 5.74), >day 6 (ES = 10.23). Day 3 > day 4 (ES = 5.06), >day 5 (ES = 6.30), >day 6 (ES = 7.61). Additionally, day 4 > day 6 (ES = 5.61). 

Zone 5 showed significant differences in day 1 > day 6 (ES = 5.51). Day 2 < day 3 (ES = −5.28). Additionally, day 3 > day 4 (ES = 3.87). 

HRmax showed significant differences in day 1 > day 6 (ES = 3.15).

#### 3.1.4. M4, In-Season

Total distance showed significant differences in day 2 > day 6 (ES = 5.05). Day 3 < day 4 (ES = −7.18), >day 5 (ES = 7.08) and >day 6 (ES = 17.68). Day 4 > day 5 (ES = 10.11), >day 6 (ES = 12.79). 

Zone 1 showed significant differences in day 3 > day 6 (ES = 9.02). Day 4 > day 6 (ES = 9.01). Day 5 > day 6 (ES = 11.40).

Zone 2 showed significant differences in day 2 > day 5 (ES = 6.40), >day 6 (ES = 5.15). Day 3 > day 5 (ES = 14.95), >day 6 (ES = 17.57). Day 4 > day 5 (ES = 11.94) and day 6 (ES = 10.43).

Zone 3 showed significant differences in day 3 < day 4 (ES = −6.97) and >day 6 (ES = 3.22). Day 4 > day 5 (ES = 10.85) and >day 6 (ES = 11.76). 

Zone 4 showed significant differences in day 2 > day 3 (ES = 8.93), >day 6 (ES = 6.34). Day 3 < day 4 (ES = −7.53). Additionally, day 4 > day 5 (ES = 5.31) and >day 6 (ES = 6.08). 

Zone 5 showed significant differences in day 2 > day 4 (ES = 10.68), >day 5 (ES = 4.36), >day 6 (ES = 7.72). Additionally, day 3 < day 5 (ES = −6.73).

HRavg showed significant differences in day 3 < day 4 (ES = −4.33) and day 4 > day 5 (ES = 4.67).

### 3.2. Comparisons between Ms

Figure 1, Figure 2 and Figure 3 displayed the differences between Ms. In Figure 1A, there were significant differences in the weekly total distance between M1 and M2 (ES = −14.72), M1 and M3 (ES = 9.26), M1 and M4 (ES = 17.42), M2 and M3 (ES = 23.09), M2 and M4 (ES = 36.22), M3 and M4 (ES = 5.93). Figure 1B showed that weekly Zone 1 presented significant differences between M1 and M2 (ES = −15.05), M2 and M3 (ES = 11.97), M2 and M4 (ES= 17.26), and M3 and M4 (ES = 6.50). Finally, Figure 1C showed that weekly Zone 2 presented significant differences between M1 and M2 (ES = 3.51), M1 and M3 (ES = 1.04), M1 and M4 (ES = 29.91), M2 and M3 (ES = 20.3), M2 and M4 (ES = 27.34).

In Figure 2A, there were significant differences for weekly Zone 3 between M1 and M2 (ES = 3.15), M1 and M3 (ES = 7.41), M1 and M4 (ES = 9.61), M2 and M3 (ES = 11.58), M2 and M4 (ES = 14.47). In Figure 2B, there were significant differences for weekly Zone 4 between M1 and M2 (ES = −12.20), M1 and M3 (ES = 6.33), M2 and M3 (ES = 13.10), M2 and M4 (ES = 24.02), M3 and M4 (ES = 7.17). Ultimately, in Figure 2C, there were significant differences for weekly Zone 5 between M1 and M2 (ES = −10.24), M1 and M3 (ES = −3.80), M2 and M3 (ES = 6.64), and M2 and M4 (ES = 9.27).

In Figure 3A, there were significant differences for HRmax between M1 and M3 (ES = 4.26) and M1 and M4 (ES = 4.92). In Figure 3B, there were significant differences for HRavg between M1 and M2 (ES = 4.00), M1 and M3 (ES = 6.50 and M1 and M4 (ES = −8.04), M2 and M3 (ES = 2.78), M2 and M4 (ES = −10.34), and M3 and M4 (ES = −11.75).

### 3.3. Comparisons between Player Positions

There were no differences between playing positions for all Ms and variables, except for Zone 4, which displayed some difference between playing positions, namely CD vs. WD (ES = −4.83) in M2 and M4 (ES = 4.96). Additionally, WD vs. WM (ES = −5.88) and WD vs. ST (ES = 5.38) in M4 (Figure 2B).

## 4. Discussion

The aim of this study was to quantify the training intensity in two Ms from pre-season and another two during in-season. The secondary aim was to analyze the influence of playing position on the intensity experienced by professional soccer players. To the best of our knowledge, this study provides the first report of daily external and internal intensity over four different Ms (two from pre-season and two from in-season) and for playing positions in Asian soccer players. This study could find significant differences in daily and accumulated intensities for the within and between match schedules.

### 4.1. Day-to-Day Intensity Variations across the Four Ms

The off-season is designed to recover, maintain and/or improve specific individual weaknesses from the in-season. However, sometimes a detraining process may occur and, in this sense, the emphasis during pre-season could be rebuilding fitness parameters after the off-season period. During the pre-season phase (M1), the training distance covered was highest on day 3, while the lowest was on day 2. The distance covered in Zone 1 was the highest on day 5, while the distance covered in Zone 2 and Zone 4 was highest on day 3. Further, the distance covered in Zone 3 was highest on day 1 while day 5 showed the highest distance covered in Zone 5. These results indicate that the high-speed running and the intensity of the training increased as the week progressed. The increase in physiological demands during the pre-season phase could be to provide optimal conditioning to the soccer players in order to prepare them for the competitive season [41]. 

There was no difference in the HRmax of the players throughout the week. The HR reported in our study was higher than that reported by Jeong et al. [12]. The HRmax was reported to be 124 ± 7 beats/min in professional Korean soccer players. This difference might be due to the external work performed by each team during their respective pre-season, individual differences in the player characteristics, and differences in the periodization plans specific to the team included in our study. This further highlights that the coaches maintained an overall consistent intensity during the entire week on the majority of the training days.

During M2, the highest training duration was on day 6. The highest distance covered was on day 3 and the lowest was on day 1. The most distance covered in Zone 1 was on day 5 and in Zone 2, 3, and 5 was on day 3. There might be a possibility that this could be due to the reduction in intensity during the start of the week and then a progressively increase as the week progressed. There was no difference in the HRmax between the days of the session, but the HRavg showed differences on days 2, 4, and 6. The HR during this period was also found to be higher than in the study by Jeong et al. [12]. It must be taken into account that training/match intensity is different today compared to more than 10 years ago.

During the in-season phase (M3 and M4), the training duration and the distance covered were reduced one day before the match. It can also be observed that the distances were increased progressively during the first three days of the week with the total distance starting to reduce from day 4. This has been associated with tapering strategies used to reduce intensity and volume on the day preceding the matches to improve player readiness [22,25,42]. Furthermore, a rest day was provided after the match during M4. These findings are in line with the previous studies in the existing literature. For instance, Kelly et al. [31] reported a recovery day after the match followed by a subsequent increase in intensity as the week progressed. It is interesting to note that the distance covered in Zone 5 was higher on M4 post-recovery day. One possible reason could be the difference in training duration, and the level of intensity applied by the coach [22]. For example, the duration of training post-match day in M4 was 73 min, whereas it was between 87–97 min during the other days except for the day before the match (day 6). This could be due to the high-intensity training approach, or simply due to the context and objectives related to the training intensity management [22]. The HRmax did not vary much between the sessions during M4. However, it was significantly different on day 1 and day 6 in M3. One possible explanation for this could be due to the increased distance covered in Zone 5 (82.5 m vs. 26.l m) on both days. The distance covered at increased speed could have contributed to the higher HR. Additionally, the training duration was higher on day 1 compared to day 6 (78.6 vs. 60.5 min). These findings reflect the day-to-day intensity variation during different Ms, and it seems that there are some marked differences in the intensity during the 1-week Ms during pre-season and in-season.

### 4.2. Comparison between Ms

The weekly total distance and the distance covered in Zone 1–5 during both Ms of the pre-season was higher compared to the in-season. Previous studies showed that pre-season intensity is generally higher compared to the in-season [12,43]. The aim during the pre-season is to work on the fitness levels of soccer players, while the aim during the in-season is focused on the technical and tactical development, and to maintain the optimal level of fitness developed during the pre-season [26]. The pre-season training period is traditionally the time when the majority of the physical preparation work is completed by players, to enable them to fulfill the physiological requirements of the competitive season [41]. Although the pre-season varies according to specific purposes of training [12], it is generally accepted that the physiological demands of this phase of training are greater than at other times during in-season. As such, maximal tests can be applied both within the gym and on-field during the pre-season phase, with the results being used to prescribe general running-based conditioning that accompanies more specific methods of training such as simulated phases of play, small-sided games, and skill-based drills [44]. There was also a significant difference in the HRmax and HRavg during both Ms of the in-season compared to the pre-season. This variation could be due to the differences in training administered during both Ms.

In particular, there seem to be differences in the total distance covered during the pre-season. The distance covered during the M2 was higher than M1. Further, the distance covered in Zone 5 was high in M2 compared to M1. One reason could be that the intensity of the training sessions could have increased and could have involved more acceleration/deceleration drills just before the beginning of the in-season. During the in-season, the total distance covered during the first M was higher when compared to the latter one. This is in agreement with the results reported by Malone et al. [26] in which soccer players covered more distance during the initial phase of the in-season compared to the final M. One further reason for this could be associated with the emphasis on physical conditioning by coaches as a continuation of the pre-season phase. The data reported within the current investigation may show that coaches adopted a pre-competition reduction in intensity to protect against injury and prevent fatigue in players as the in-season progresses. The data from our study suggested that training intensity is periodically manipulated before the competition, in order to reduce the risk of injury and improve potential match performance as the competition progresses [45,46,47]. 

### 4.3. Comparisons between Player Positions

Our results revealed that there were significant differences in the distance covered in Zone 4, it varied significantly between CD, WD, WM and ST during M2 and M4. The WD was found to cover the most distance in Zone 4 during both Ms. Wide areas are generally less congested compared to the central ones, resulting in increased chances of achieving high speed [48]. Further, since CD operated in congested, and central areas of the pitch, this could have led them to cover less distance in high-speed running when compared to WD [49]. Previous research by Abbott et al. [48] also indicated that the high-speed running and sprinting distances covered by WD and WA are similar because they operate on the flanks of the pitch. However, our results contradict these findings as there was a significant difference observed between both playing positions during M4. When comparing WD and ST, the lesser involvement of the ST during defensive tasks could explain the lesser distance covered in high-intensity zones [48]. In addition, every position saw similar efforts during the other periods of the M and there was no significant difference in the distance covered in the other M.

### 4.4. Practical Applications, Limitations and Considerations for Future Studies

This study provides useful information regarding the intensity employed by professional soccer teams during the pre- and in-season. It provides further evidence of the value of using the combination of different measures of intensity to fully evaluate the patterns observed across the two periods of the season. For coaches and practitioners, the study generates reference values for professional players, which can be considered when planning training sessions, and also designing and implementing training programs considering the physiological demands of each playing position. Considering the importance of summarizing the main evidence about the influence of factors for optimizing the exercises and the potential practical applications for coaches, it is suggested that HR should be used with some parsimony in controlling the intensity of the training. In maximizing the physical fitness of players from different field positions, coaches can take into account that only speed zones higher than 14.4 km·h^−1^ denoted some significant differences for player positions.

Despite the previous practical applications, this study is based on one top professional Asian soccer team with only a small sample size and for those reasons, it cannot represent the usual training demands of other leagues/countries. Thus, future studies with larger sample sizes and from other leagues/countries should be considered. Moreover, playing position analysis was limited to the small number of participants in each position (2–6 players). Furthermore, it was previously suggested to conduct future studies, that included players that also completed exercise training sessions, without competitive matches [22]. Even so, we consider that data are still limited because there are other variables such as acceleration, deceleration, player load, and metabolic power as well as technical, tactical, or other physical and physiological variables or information about training content and periodization structure that could also give important knowledge and references for training which makes us suggest the use of covariates in future studies. In addition, future studies can consider training and match data to amplify science and confirm these results. Additionally, it is important to understand whether several contextual variables, such as the opponent level, the match result, or the time during the season, could affect the results. Lastly, the higher running speed zone included a threshold higher than 19.8 km·h^−1^ and future studies may want to consider other speed thresholds such as higher, (i.e., 24 km·h^−1^) to provide additional information. However, in the present study, several players did not achieve such running speed as previously explained. This information should be considered by performance and technical coaches, especially those from the Iranian league.

## 5. Conclusions

This study quantifies the daily soccer training and accumulative weekly intensity of a team from the Asian Professional League in four different Ms schedules. It is important to note that customary external intensity did not exhibit a regular pattern for the analyzed Ms. However, there was a similar tendency of constant HRmax (range values: 165–188 bpm’s) and HRavg (range values: 119–145 values) over the different Ms. Moreover, this study confirmed previous evidence that there was a general tendency to decrease intensity in the last day of the M.

In general, there was a higher external intensity in Ms from the pre-season than in-season. Finally, only speed zones higher than 14.4 km·h^−1^ denoted some significant differences for player positions. WD and WM showed higher distances covered with high intensity. In opposition, internal intensity does not vary from different playing positions.

## Figures and Tables

**Figure 1 healthcare-10-01367-f001:**
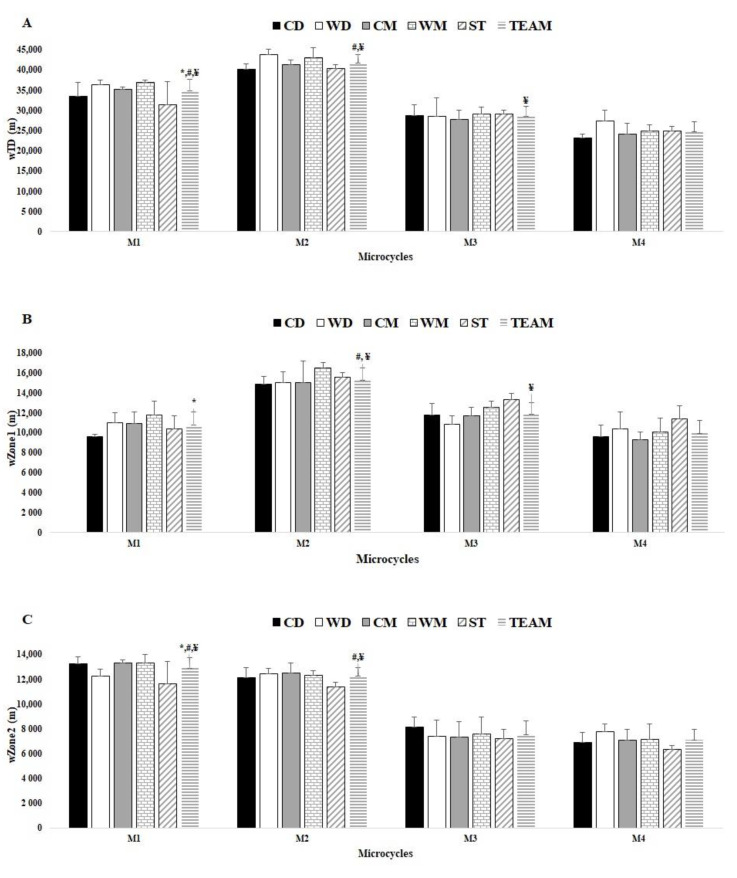
Comparisons between player positions for weekly total distance (wTD), Zone 1 (wZone1), and Zone 2 (wZone2) for each microcycle, (e.g., M1, M2, M3, and M4) and between Ms by team average. (**A**) wTD; (**B**) wZone1; (**C**) wZone2; * Denotes difference from M2; # denotes difference from M3; ¥ denotes difference from M4; all *p* < 0.05.

**Figure 2 healthcare-10-01367-f002:**
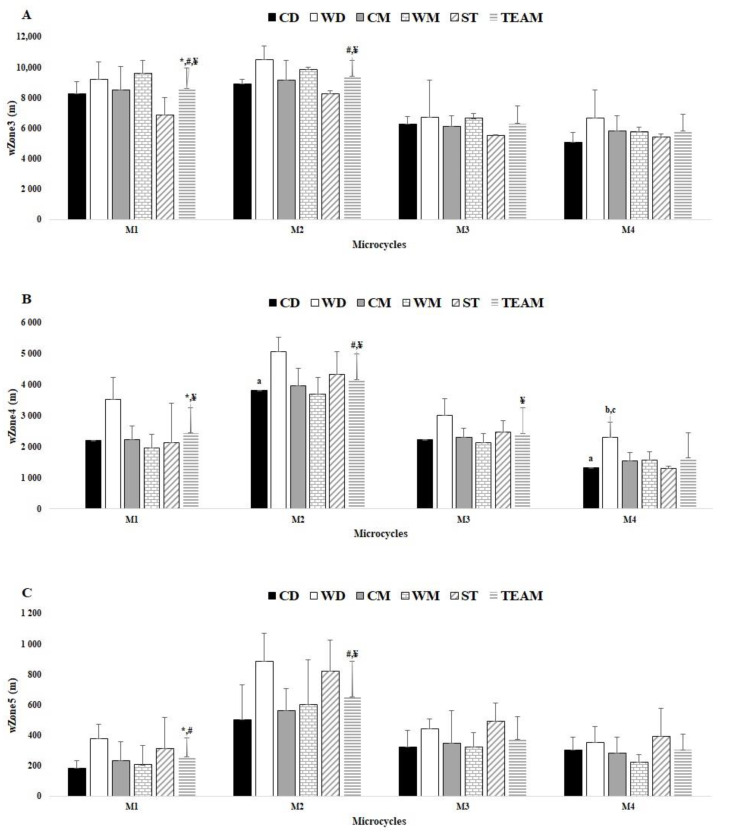
Comparisons between player positions for weekly Zones 3, 4, and 5 for each microcycle, (e.g., M1, M2, M3, and M4) and between Ms by team average. (**A**) wZone3; (**B**) wZone4; (**C**) wZone5; a denotes difference from WD, b denotes difference from WM, c denotes difference from ST, * denotes difference from M2, # denotes difference from M3, ¥ denotes difference from M4, all *p* < 0.05.

**Figure 3 healthcare-10-01367-f003:**
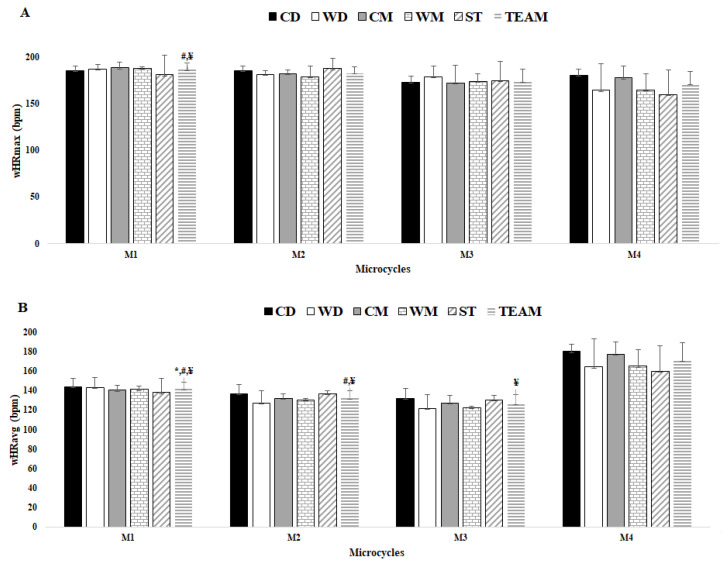
Comparisons between player positions for average HRmax and average HRavg for each microcycle, (e.g., M1, M2, M3, and M4) and between Ms by team average. (**A**) wHRmax; (**B**) wHRavg; * Denotes difference from M2; # denotes difference from M3; ¥ denotes difference from M4; all *p* < 0.05.

**Table 1 healthcare-10-01367-t001:** Weekly training characterization of Ms from pre- and in-season.

**M1, Pre-Season**						
Day 1	Day 2	Day 3	Day 4	Day 5	Day 6	Day 7
training sessions					day-off	
**M2, Pre-Season**						
Day 1	Day 2	Day 3	Day 4	Day 5	Day 6	Day 7
training sessions						day-off
**M3, In-Season**						
Day 1	Day 2	Day 3	Day 4	Day 5	Day 6	Day 7
training sessions						match-day
**M4, In-Season**						
Day 1	Day 2	Day 3	Day 4	Day 5	Day 6	Day 7
day-off	training sessions					match-day

**Table 2 healthcare-10-01367-t002:** Training duration (minutes) during the 7-day testing period for squad average.

Periods	Day 1	Day 2	Day 3	Day 4	Day 5	Day 6	Day 7
**M1, pre-season**	83.3 ± 0.0 ^a,b,c,d^	76.6 ± 2.0 ^b,c,d^	93.7 ± 0.4 ^c,d^	95.3 ± 1.9	92.4 ± 1.0	X	X
**M2, pre-season**	70.8 ± 3.1 ^a,b,c,d,e^	98.9 ± 1.0 ^b,e^	107.2 ± 0.0 ^c,d,e^	98.7 ± 1.6 ^e^	100.9 ± 0.0 ^e^	110.3 ± 0.0	X
**M3, in-season**	78.6 ± 3.3 ^a,e^	97.6 ± 0.0 ^b,c,d,e^	80.4 ± 0.1 ^c,e^	73.4 ± 0.0 ^d,e^	80.8 ± 0.5 ^e^	60.5 ± 0.1	MD
**M4, in-season**	X	73.1 ± 5.9 ^b^	97.0 ± 0.0 ^e^	91.4 ± 2.9 ^e^	87.2 ± 0.9 ^e^	60.1 ± 0.0	MD

M, microcycle; X, day-off; MD = match-day; ^a^, denotes difference from day 2; ^b^, denotes difference from day 3; ^c^, denotes difference from day 4; ^d^, denotes difference from day 5; ^e^, denotes difference from day 6; all *p* < 0.05.

**Table 3 healthcare-10-01367-t003:** Distances covered at different speed thresholds and heart rate variables (representative of squad average data) during the 7-day period in pre-season.

**M1, Pre-Season**	**Day 1**	**Day 2**	**Day 3**	**Day 4**	**Day 5**	**Day 6**	**Day 7**
**Total Distance (m)**	6951.5 ± 143.8 ^a,b^	6000.1 ± 153.4 ^b,d^	8135.8 ± 196.8 ^c,d^	6422.4 ± 217.9 ^d^	7117.0 ± 191.5	X	X
**Zone1 (m)**	1671.8 ± 109.1 ^b,c,d^	1933.8 ± 104.4 ^c,d^	2233.1 ± 72.2 ^d^	2349.2 ± 63.0	2538.4 ± 66.0	X	X
**Zone2 (m)**	2600.1 ± 49.5 ^a,b^	2155.6 ± 89.8 ^b,d^	3174.4 ± 80.2 ^c,d^	2224.1 ± 116.0	2587.8 ± 66.0	X	X
**Zone3 (m)**	2131.2 ± 83.0 ^a,c,d^	1504.4 ± 79.6 ^b^	2041.3 ± 108.8 ^c,d^	1279.6 ± 60.8 ^d^	1530.5 ± 88.3	X	X
**Zone4 (m)**	519.4 ± 53.0 ^a^	359.8 ± 35.6 ^b,c^	619.9 ± 31.2 ^d^	524.1 ± 38.2 ^d^	384.9 ± 45.0	X	X
**Zone5 (m)**	29.1 ± 3.4	46.5 ± 7.1	67.2 ± 10.7	45.3 ± 5.1	75.4 ± 16.9	X	X
**HRmax (bpm)**	187 ± 2	184 ± 4	188 ± 2	186 ± 2	186 ± 3	X	X
**HRavg (bpm)**	145 ± 3 ^d^	141 ± 3	144 ± 1 ^d^	138 ± 2	139 ± 2	X	X
**M2, Pre-Season**	Day 1	Day 2	Day 3	Day 4	Day 5	Day 6	Day 7
**Total Distance (m)**	5734.5 ± 216.4 ^b,d^	6807.0 ± 223.1 ^b,d^	8742.1 ± 136.2 ^c,e^	6215.3 ± 150.7 ^d^	8091.8 ± 250.6 ^e^	6133.6 ± 142.5	X
**Zone1 (m)**	1999.4 ± 80.3 ^a,b,d,e^	2608.2 ± 77.5 ^b,c^	2910.2 ± 63.6 ^c^	2285.6 ± 54.3 ^d,e^	2914.3 ± 105.2	2670.5 ± 91.3	X
**Zone2 (m)**	1650.2 ± 98.6 ^b,d^	1747.0 ± 61.3 ^b,c,d^	2665.3 ± 80.8 ^c,e^	2097.0 ± 72.6 ^e^	2330.6 ± 94.8 ^e^	1675.6 ± 36.4	X
**Zone3 (m)**	1143.9 ± 93.8 ^a,b,d^	1760.7 ± 91.3 ^b^	2097.0 ± 75.8 ^c,d,e^	1377.9 ± 75.1	1554.5 ± 65.9	1387.8 ± 67.9	X
**Zone4 (m)**	809.8 ± 86.1 ^c,d,e^	618.4 ± 37.1 ^b,c,d,e^	762.3 ± 44.6 ^c,d,e^	421.7 ± 34.4 ^d^	1201.9 ± 6.8 ^e^	358.9 ± 28.3	X
**Zone5 (m)**	131.1 ± 22.3 ^b,c,e^	72.8 ± 12.9 ^b^	307.2 ± 19.3 ^c,d,e^	30.1 ± 5.2	90.5 ± 21.7	40.7 ± 4.8	X
**HRmax (bpm)**	170 ± 9	188 ± 2	187 ± 2	176 ± 5	186 ± 2 ^e^	180 ± 2	X
**HRavg (bpm)**	128 ± 7	138 ± 3 ^c,e^	139 ± 3 ^c,e^	127 ± 3	135 ± 2 ^e^	124 ± 2	X

M, microcycle; MD, match-day; bpm, beats per minute; m, meters; HRmax, heart rate maximum; HRavg, heart rate average; X, Day Off; ^a^, denotes difference from day 2; ^b^, denotes difference from day 3; ^c^, denotes difference from day 4; ^d^, denotes difference from day 5; ^e^, denotes difference from day 6; all *p* < 0.05.

**Table 4 healthcare-10-01367-t004:** Distances covered at different speed thresholds and heart rate variables (representative of squad average data) during the 7-day period in season.

**M3, In-Season**	**Day 1**	**Day 2**	**Day 3**	**Day 4**	**Day 5**	**Day 6**	**Day 7**
**Total Distance (m)**	4818.1 ± 370.9 ^b^	5488.8 ± 189.1 ^c,d,e^	6966.4 ± 415.3 ^c,d,e^	4019.6 ± 96.3 ^e^	3757.0 ± 174.7	3584.6 ± 60.0	MD
**Zone1 (m)**	1791.0 ± 129.0 ^a,b^	2394.5 ± 83.6 ^c,d,e^	2675.6 ± 140.8 ^c,d,e^	1750.3 ± 38.2 ^e^	1798.1 ± 126.4	1616.9 ± 35.9	MD
**Zone2 (m)**	1423.4 ± 178.0	1669.6 ± 91.4 ^c,d,e^	1583.6 ± 98.2 ^c,d,e^	904.6 ± 21.3 ^e^	892.3 ± 50.2 ^e^	1061.7 ± 20.5	MD
**Zone3 (m)**	1115.9 ± 142.6 ^b^	937.9 ± 78.7 ^b,e^	1953.5 ± 154.1 ^c,d,e^	942.1 ± 37.7 ^d,e^	682.6 ± 28.3	630.7 ± 31.4	MD
**Zone4 (m)**	405.3 ± 50.3	445.0 ± 25.8 ^b,c,d,e^	651.4 ± 74.3 ^c,d,e^	366.7 ± 28.5 ^e^	305.7 ± 22.6	249.0 ± 8.3	MD
**Zone5 (m)**	82.5 ± 14.1 ^e^	41.6 ± 9.2 ^b^	102.3 ± 13.4 ^c^	55.9 ± 10.4	78.4 ± 19.7	26.1 ± 3.3	MD
**HRmax (bpm)**	180 ± 3 ^e^	174 ± 5	179 ± 6	172 ± 4	189 ± 2	167 ± 5	MD
**HRavg (bpm)**	129 ± 4	125 ± 3	139 ± 6	126 ± 2	121 ± 2	124 ± 2	MD
**M4, In-Season**	Day 1	Day 2	Day 3	Day 4	Day 5	Day 6	Day 7
**Total Distance (m)**	X	5739.9 ± 628.8 ^e^	4883.3 ± 96.2 ^c,d,e^	6768.5 ± 358.5 ^d,e^	3988.4 ± 150.7	3486.8 ± 56.8	MD
**Zone1 (m)**	X	2042.9 ± 245.8	2091.3 ± 85.1 ^e^	2424.9 ± 139.8 ^e^	2085.9 ± 63.5 ^e^	1509.6 ± 32.9	MD
**Zone2 (m)**	X	1867.4 ± 228.1 ^d,e^	1535.6 ± 23.4 ^d,e^	1821.0 ± 102.2 ^d,e^	792.4 ± 66.3	1027.8 ± 33.5	MD
**Zone3 (m)**	X	1299.7 ± 222.4	1043.8 ± 115.5 ^c,e^	1989.5 ± 153.2 ^d,e^	758.9 ± 47.7	661.2 ± 45.3	MD
**Zone4 (m)**	X	396.0 ± 26.8 ^b,e^	193.6 ± 17.6 ^c^	493.8 ± 53.6 ^d,e^	269.8 ± 26.3	249.9 ± 18.5	MD
**Zone5 (m)**	X	133.9 ± 11.4 ^c,d,e^	18.9 ± 3.3 ^d^	39.3 ± 5.2	81.3 ± 12.7	38.3 ± 13.3	MD
**HRmax (bpm)**	X	170 ± 6	166 ± 7	176 ± 5	170 ± 4	165 ± 8	MD
**HRavg (bpm)**	X	127 ± 4	120 ± 3 ^c^	133 ± 3 ^d^	119 ± 3	121 ± 5	MD

M, microcycle; MD, match-day; bpm, beats per minute; m, meters; HRmax, heart rate maximum; HRavg, heart rate average; X, Day Off; ^a^, denotes difference from day 2; ^b^, denotes difference from day 3; ^c^, denotes difference from day 4; ^d^, denotes difference from day 5; ^e^, denotes difference from day 6; all *p* < 0.05.

## Data Availability

The data presented in this study are available on request from the corresponding author.

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
