# Peer review of "Quantification of Pre-Season and In-Season Training Intensity across an Entire Competitive Season of Asian Professional Soccer Players"

_healthcare, 2022, doi:10.3390/healthcare10081367_

Round 1
Reviewer 1 Report
Abstract and Introduction
Minor grammar and spelling edit required.
Edit spacing and punctuation throughout.
Ensure most contemporary literature are referenced, for example line 47 - reducing injury risk in team sports might be appropriate to reference Gabbett's work?
Line 49-52 - from a practitioners perspective and the research referenced [12] that only compared one week of preseason versus one week of in-season. At the end of pre-season, players are not at maximum fitness capacity so to 'maintain' the pre-season fitness is not a true reflection of in-season training aims. maybe amend this statement to reflect.
Line 89-92 - S&C coaches or performance coaches and technical coaches? by definition and practical roles all coaches will need this information, as it won't be S&C coaches that 'manipulate training volume'. can this be amended?
Methods
Minor grammar and spelling edit required.
was zone 5 classified as high-intensity and sprint? is there a reason why these two significant metrics were combined? please explain. considering the notion surrounding injury risk and sprint exposure. maybe a zone 6 might highlight some significant differences? please consider
as the paper address differences in training intensity during pre-season and in-season. maybe some contextual information around the training content and periodisation structure would be interesting? was there a planned tactical and physical difference between the daily sessions?
L113 - can the standardised 30min be explained in context of the session? did these change daily?
Results
L178 - capital T for Table 2 and 3. please review throughout paper
Do the journal guidelines allows abbreviations to start a sentence? if not, maybe amend throughout paper?
Discussion
is there a reason why '2' is used to describe the Ms, while 'four' is used to describe them accumulatively, L419-423 as example. please standardise throughout paper.
In elite football, the off-season is designed to recovery and maintain and/or improve specific individual weaknesses so the de-training is often minimal. maybe a re-phrase of terminology.
maybe expand the practical applications for coaches based on the new findings
Please review the use of 'the' throughout paper - for example L283
Reviewer 2 Report
I would like to express my gratitude regarding the opportunity to review this manuscript.
It is an interesting study, congratulations, but at this stage still requiring improvements. Below suggestions with line indication:
6 – “Motor Performance” (with uppercase – suggested).
6-16 – Please include authors´ initials close to the emails.
23 – Please consider “HR” abbreviation (repeated in the abstract).
24 – “M” not previously abbreviated (Microcycle?), please consider in full in the first appearance in the text.
31-32 – Please consider reducing the keywords aiming directing the readers to the study topic.
40 – “25.1 km/h-1” – Units not close to values, different format comparting to abstract. Please review and standardize throughout the manuscript.
35-58 – Please consider reducing the paragraph size.
59 – Please review if not more than one space after end point.
80 – “Ms” – First time in the text should be in full.
86 - studies available or “available studies”? Please carefully review the English throughout the manuscript.
99 – “microcycles” – Abbreviation suggested.
104 – “Asian professional soccer league” – Previously in uppercase. Please standardize throughout the manuscript.
112 – “Data from rehabilitation or recuperation was excluded”? This sentence is not clear. Any relation with inclusion and exclusion criteria? Please describe.
120 – Some titles in the table in bold and others not, please standardize.
123 – Please place the playing positions in full in the first appearance in the text. Some readers of the article may not be totally familiarized with soccer.
122-129 – Please describe in detail the inclusion and exclusion criteria.
150 – Please consider using km·h-1 instead of km.h-1 in all text (different dot format – middle dot).
152 – Why were these running speed used? Please consider providing reference.
155 – “RED” – Please Review
160 – “))” – Please correct.
165 – Please describe statistical power.
122-129 – Please describe players characteristics (age, height, weight, experience, and others).
130 – Please describe the procedures in detail. Type of grass, data collection time of day (circadian effect), players nutrition? Who collected the data, training and experience? All details should be considered and detailed in the text.
206-229 – Please carefully review the results presented in text. The effect size categorization relevant and not presented (only numeric). Maybe a chart could provide readers with a more direct understanding. Please consider this and other possibilities in all the results section, currently with very long text and data in tables (results section with ± 6 pages). All the results section format should be considered, to provide readers with the best reading and interpretation conditions.
233 – Please consider placing the values of the table in the same line to provide readers better interpretation conditions.
419 – “mesocycles” – not necessary, previously abbreviated.
434 – “high-speed running” – HSR suggested in all manuscript (first time in full and afterward abbreviated).
536 – Limitations development is suggested. This data is related to a specific team, in a specific country and context/level. Also, some playing positions present very few players to extrapolate the results to other soccer clubs. These are only some examples that can be explored in limitations.
542 - Please consider directing the text of the conclusions section more towards objective findings/ideas and take-home messages with practical application.
576 – Please carefully review references according to journal instructions for authors (for example articles titles in upper and lowercase – standardization suggested).
Reviewer 3 Report
The part that worries me is the one related to the statistics, although the methods are correct, however, in the part that talks about the size of Hedge's g effect, they must indicate that the criterion to classify is in absolute value.
The selection of the weeks was based on similar characteristics but they do not indicate what they were and why those characteristics were selected.
The authors considered fitting a model other than the repeated measures ANOVA that would allow the effect of several covariates to be studied at the same time, or a repeated measures ANOVA with several factors was used.
Round 2
Reviewer 2 Report
Dear authors,
Thank you for considering my suggestions and incorporating them into the manuscript. Below some more small suggestions regarding details, with line indication.
7 & 9 - End of line is different compared to others (e.g. 14 “;”). Please standardize according to journal instructions for authors and journals´ template.
66 - Please correct after [17]
107 - “Ms” suggested. Please carefully review all the manuscript regarding the small details like this.
108 - “Asian Professional League” in line 89. Please review and standardize throughout the manuscript (same for example in line 130).
109 - Please confirm if not “Ms” instead of “M”.
124 – “i.e. M1 and M1 124 and M4 had five training sessions while M2 and M3 had four training sessions”. – I believe there is a mistake here (M1 + M1). Please also do not forget to close parentheses.
128 - “Ms”?
134 – “body mass index of 22.5 ± 1.1 kg/m2.” – Before “and” suggested.
139 & 117 – “recovery sessions” – suggestion.
144 – “sessions” – suggestion.
146 – Please GPS in full in the first appearance in the text.
167 – “[22, 34]” – Without space in all manuscript citations with more than one number, please correct.
170 - Please close parentheses.
172 - “Ms”?
175 – “red” in lowercase suggested.
201 - “Ms”?
261 – Please confirm if not more than one space in before “Also”.
347 & 400 – “(M)” may be confused to readers because not in figures, only M1, M2, …. no abbreviation is suggested in the legend.
587 & 592 - “Ms”?
588 – “Ms analysed” or “analyzed Ms”? Please carefully review the English in this line and throughout the entire manuscript.
601 – “O..;” – Please correct. In line 708 & 713 the same. Please carefully review all the references.
Congratulations for the research. Keep up the good work.
Reviewer 3 Report
As a suggestion for future work, try to work with larger sample sizes and evaluate covariates that may influence the results.
